# Krill Oil Attenuates Cognitive Impairment by the Regulation of Oxidative Stress and Neuronal Apoptosis in an Amyloid β-Induced Alzheimer’s Disease Mouse Model

**DOI:** 10.3390/molecules25173942

**Published:** 2020-08-28

**Authors:** Ji Hyun Kim, Hui Wen Meng, Mei Tong He, Ji Myung Choi, Dongjun Lee, Eun Ju Cho

**Affiliations:** 1Department of Food Science and Nutrition, Pusan National University, Busan 46241, Korea; kjjjjhh11@naver.com (J.H.K.); 791386767mhw@gmail.com (H.W.M.); skyham16@gmail.com (M.T.H.); poutia@naver.com (J.M.C.); 2Department of Food and Nutrition, Kyungsung University, Busan 48434, Korea; 3NZ Origin Ltd., Busan 48059, Korea; djlee@nzorigin.com

**Keywords:** Alzheimer’s disease, amyloid beta, cognition, memory, krill oil, oxidative stress

## Abstract

In the present study, we investigated the cognitive improvement effects and its mechanisms of krill oil (KO) in Aβ_25–35_-induced Alzheimer’s disease (AD) mouse model. The Aβ_25–35_-injected AD mouse showed memory and cognitive impairment in the behavior tests. However, the administration of KO improved novel object recognition ability and passive avoidance ability compared with Aβ_25–35_-injected control mice in behavior tests. In addition, KO-administered mice showed shorter latency to find the hidden platform in a Morris water maze test, indicating that KO improved learning and memory abilities. To evaluate the cognitive improvement mechanisms of KO, we measured the oxidative stress-related biomarkers and apoptosis-related protein expressions in the brain. The administration of KO inhibited oxidative stress-related biomarkers such as reactive oxygen species, malondialdehyde, and nitric oxide compared with AD control mice induced by Aβ_25–35_. In addition, KO-administered mice showed down-regulation of Bax/Bcl-2 ratio in the brain. Therefore, this study indicated that KO-administered mice improved cognitive function against Aβ_25–35_ by attenuations of neuronal oxidative stress and neuronal apoptosis. It suggests that KO might be a potential agent for prevention and treatment of AD.

## 1. Introduction

Amyloid beta peptide (Aβ) is a major pathological feature in Alzheimer’s disease (AD), and it is widely accepted as a cause of AD [1]. Accumulations of Aβ induce neuronal oxidative stress through over-productions of reactive oxygen species (ROS) and reactive nitrogen species (RNS) in the brain [2]. In addition, neuronal oxidative stress leads to neuroinflammation, mitochondrial dysfunction, and DNA/RNA damages in the brain [2,3]. In particular, neuronal oxidative stress induced by accumulation of Aβ leads to neuronal cell death by stimuli of the apoptotic signaling [4]. The Aβ inhibited anti-apoptotic factor such as B-cell lymphoma 2 (Bcl-2) and stimulated the pro-apoptotic factor such as Bcl-2-associated X protein (Bax) [5]. Therefore, Aβ leads to neuronal oxidative stress and neuronal apoptosis, resulting in cognitive impairment and memory deficit. In the treatment of AD, donepezil has been approved by Food and Drug Administration and widely prescribed for AD patients [6]. However, donepezil has several clinical adverse effects such as dizziness, fatigue, and vomiting in humans [6,7]. Therefore, many studies were focused on cognitive improvement effect and its mechanisms of natural products that can contribute the prevention and treatment of AD without side effects [8,9].

Krill oil (KO) is extracted from Antarctic krill (*Euphausia superba*) which is a small marine crustacean found in the Antarctic Ocean [10]. The KO contains essential n-3 polyunsaturated fatty acids (PUFAs) such as docosahextanoic acid (DHA) and eicosapentaenoic acid (EPA) [10,11]. In addition, n-3 PUFAs in KO are incorporated into phospholipids, mainly phosphatidylcholine (PC), which increases the bioavailability with higher absorption rate than fish oil [12]. Astaxanthin is a marine-derived natural antioxidant also mainly found in KO [13]. Therefore, these characteristics of KO such as high contents of n-3 PUFAs, PC, and astaxanthin suggest various health beneficial activities. Previous studies have reported health benefits of KO such as anti-hyperlipidemia, anti-inflammation, and anti-arthritis effects [14,15,16]. Several studies also demonstrated that neuroprotective and cognitive improvement effect of KO. The administration of KO improved cognitive function in d-galactose-induced cognitive impairment mice and age-related SAMP8 mice by regulation of oxidative stress, proteomic changes, and Aβ deposition [17,18]. However, the protective effect of KO on cognitive impairment by regulation of neuronal oxidative stress and neuronal apoptosis in Aβ-induced AD mouse model has not been reported.

Therefore, this study investigated the cognitive improvement effect of KO by behavior tests such as novel object recognition test, passive avoidance test, and Morris water maze test in Aβ_25–35_-induced AD mouse model. In addition, to evaluate the cognitive improvement mechanisms of KO, we measured the oxidative stress and neuronal apoptosis-related biomarkers in the brain.

## 2. Results

### 2.1. Effect of Krill Oil (KO) on Oxidative Stress in Aβ-Induced AD Mouse

To examine the mechanisms of KO on cognitive function, we measured the oxidative stress-related biomarkers such as ROS, malondialdehyde (MDA), and nitric oxide (NO) in the brain tissue. As shown in Figure 1A, Aβ_25–35_-injected control group significantly elevated the ROS production, compared with normal group. However, KO- and donepezil-administered groups significantly inhibited the ROS production, compared with control group.

Figure 1B shows the MDA levels in the brains of Aβ_25–35_-induced mice. The MDA levels of Aβ_25–35_-injected control group was 107.41 nmol/mg protein, while in the normal group the MDA levels were significantly decreased to 81.35 nmol/mg protein. However, the KO at concentrations of 100, 200 and 500 significantly inhibited the MDA levels to 84.79, 86.53 and 75.93 nmol/mg protein compared with control group. In addition, donepezil-administered group also decreased the MDA level to 83.91 nmol/mg protein in the brain.

As shown in Figure 1C, the NO levels of control group injected by Aβ_25–35_ was 15.24 μmol/L/mg protein, whereas mice of normal group significantly reduced the NO levels to 8.66 μmol/L/mg protein in the brain. However, administrations of KO at concentration of 100, 200 and 500 significantly inhibited the MDA levels to 5.55, 4.01 and 5.66 μmol/L/mg protein compared with control group. In addition, donepezil-administered group also inhibited the NO level to 6.57 μmol/L/mg protein in the brain. Therefore, these results suggest that KO inhibited the neuronal oxidative stress by decreases of ROS, MDA, and NO in the brain of Aβ_25–35_-injected AD mouse.

### 2.2. Effect of Krill Oil (KO) on Novel Object Recognition Ability

The effect of KO on object cognitive ability in Aβ_25–35_-induced cognitive impairment mouse model was shown in Figure 2. On the first day (training day), all experimental animals freely explored two identical objects (A, A’). None of the groups showed a significant difference of the percentage of time exploring of between two identical objects (about 50-50), and equivalently explored objects (Figure 2A). In the test day, one of the two objects was replaced with a novel object (A, B). The Aβ_25–35_-non-injected normal group showed higher exploration of novel object than that of the familiar object, while Aβ_25–35_-injected control group showed non-significance between familiar and novel objects (Figure 2B). The percentage of time exploring of the familiar and novel object in Aβ_25–35_-injected control group was 51.31% and 48.69%, respectively.

These results show no significant difference in exploration between familiar and novel objects. However, the Aβ_25–35_-non-injected normal group showed that exploration of novel object is higher than that of the familiar object, indicating significant difference in exploration of familiar and novel object. In addition, KO at various dose and donepezil-administered groups significantly increased exploration of novel object, similar to normal group. Therefore, this result suggests that the administration of KO can attenuate Aβ_25–35_-induced object recognition deficit.

### 2.3. Effect of Krill Oil (KO) on Passive Avoidance Ability

As shown in Figure 3, we investigated the effect of KO on passive avoidance ability in Aβ_25–35_-induced AD mouse. In the training day, an entry latency time in the passive avoidance test showed no significance among all experimental groups. However, in the retention trial, Aβ_25–35_-injected control group showed significant decrease of the latency time to enter the brightly chamber, compared with normal group. In addition, mice-administered with KO and donepezil significantly increased the latency time to enter the brightly chamber, compared with control group. These findings indicated that administration of KO attenuated passive avoidance ability in Aβ_25–35_-induced AD mouse model.

### 2.4. Effect of Krill Oil (KO) on Learning and Memory Ability in Morris Water Maze Test

To evaluate the learning and memory abilities of KO against Aβ_25–35_-induced cognitive impairment, we carried out a Morris water maze test. As shown in Figure 4, Aβ_25–35_-injected mice of control group exhibited a significantly longer latency to reach hidden platform, compared with normal group of mice. In contrast, administration of KO and donepezil in Aβ_25–35_-induced mice indicated a significantly shorter latency to find hidden platform, compared with control group. Figure 5 indicated the % time spent in the platform-located target quadrant according to other previous studies. The normal group stayed target quadrant during 35.26 ± 5.70% (21.16 s of 60 s), while control group stayed target quadrant during 26.47 ± 1.84% (15.88 s of 60 s). In addition, KO100, KO200, KO500, and donepezil groups showed the percentage of target quadrant without non-calculated relative normal group, 33.87 ± 3.46%, 31.05 ± 5.26%, 35.90 ± 5.18% and 31.21 ± 5.61%, respectively. Figure 6 exhibited the escape latency time to find the hidden or exposed platform. The latency time to reach the hidden platform showed significant difference among groups, increase of Aβ_25–35_-injected control group and decrease of non-injected normal group. However, the mice of all groups showed no significant differences were observed in the exposed platform, indicating that visual or physical activities were not related to find the platform.

### 2.5. Effect of Krill Oil (KO) on Neuronal Apoptosis in Aβ-Induced AD Mouse

To investigate the effect of KO on neuronal apoptosis in Aβ_25–35_-induced AD mouse, we measured the protein expressions of Bax and Bcl-2 in the brain (Figure 7). The Aβ_25–35_-injected control group significantly increased the ratio of Bax/Bcl-2, compared with normal group. However, the KO-administered group such as KO100, KO200 and KO500 significantly down-regulated the ratio of Bax/Bcl-2 proteins. In addition, donepezil-administered group also down-regulated the ratio of Bax/Bcl-2 proteins. Therefore, this result indicated that KO attenuated the neuronal apoptosis by down-regulation of Bax/Bcl-2 ratio in the brain.

## 3. Discussion

AD, an age-related neurodegenerative disorder, is caused by accumulation of Aβ. Aβ is produced by amyloidogenic pathway, which is processing of amyloid precursor protein (APP) to Aβ via activation of two enzymes such as β- and γ-secretase [1]. In addition, Aβ-injected mice display several AD-like pathological symptoms such as the neuronal oxidative stress and neuronal apoptosis in the brain tissue [19,20]. Drugs such as donepezil used in AD patients for treatment of AD, attenuate the cognitive impairment by regulation of the cholinesterase inhibition in the brain [6]. However, donepezil could potentially delay the clinical symptoms of AD, but it did not show beneficial effect in one-third of AD patients and exhibited side effects such as gastrointestinal, fatigue, and muscle cramps in human [6,7]. Therefore, recently a number of studies have approached the development of effective natural agents for the prevention and treatment of AD without side effects [21].

The n-3 PUFAs such as EPA and DHA are mainly found in the marine materials such as fish and fish oil. The DHA is contained in neuron membrane and low amounts in AD patients were observed [22,23]. The supplementations of EPA and DHA improved cognitive dysfunction by anti-oxidant, anti-inflammation, and degradation of Aβ plaque in AD patients [23]. In addition, the brain lacks the ability to synthesize DHA, thereby plant-derived n-3 PUFAs are less efficient than marine source-derived n-3 PUFAs in the body [24]. Therefore, dietary consumption of n-3 PUFAs derived from marine sources such as KO is crucial to improve cognitive function. In addition, KO contains high amount of the phospholipids, especially PC. Dietary supplementation of PC improved cognitive function by inhibition of oxidative stress and Aβ-induced neurotoxicity [25,26]. The administration of astaxantin, which is an active component of KO, improved cognitive and memory function by inhibition of oxidative stress and Aβ plaque level in an Aβ-treated rat model [27]. However, the cognitive improvement effects and its mechanisms of KO, which is rich sources of PUFAs, PC, and astaxantin, against Aβ_25–35_-induced cognitive impairment have not been investigated yet. In the present study, we evaluated the effect of KO on cognitive impairment in an Aβ_25–35_ induced- AD mouse model.

We investigated the cognitive improvement effects of KO using behavior tests such as the novel object recognition test, passive avoidance test, and Morris water maze test, which are widely used to evaluate the cognitive function in Aβ_25–35_-induced AD mice [19,28]. In addition, many studies reported that behavior tests were well established for evaluation of cognitive improvement effects by administration of natural products in AD mice [19,29]. The novel object recognition test based on the fact that mice have a tendency to explore more a novel object than a familiar object for assessing recognition memory [30]. The normal mice explore novel object longer time, Aβ_25–35_-induced cognitive impairment mice showed no difference between length of the novel and familiar objects [30,31]. In this study, Aβ-injected AD mice exhibited cognitive impairments, but administration of KO significantly attenuated the cognitive impairments in the novel object recognition test.

Passive avoidance test is based on tendency of mice to natural preference toward dark environment [32]. The Aβ_25–35_-induced AD mice more frequently enter the electric shock-consisted the dark chamber than electric shock-non-consisted light chamber by damage of passive avoidance ability, compared with normal mouse [20,33]. Our result showed that Aβ_25–35_-induced AD mice disrupted passive avoidance ability. However, administrations of KO and donepezil in AD mouse improved passive avoidance ability by higher entrance to the light chamber. In particular, similarly our result, previous study investigated that administration of donepezil increased passive avoidance ability in cognitive impairment mice model. Therefore, these findings indicated improvement effects of KO on passive avoidance in Aβ_25–35_-induced AD mouse.

To evaluate the learning and memory abilities of KO, we carried out a Morris water maze test. In this result, the injection of Aβ_25–35_ deteriorated the learning and memory abilities by showing longer time to find latency the hidden platform. However, administration of KO showed shorter time to find latency the hidden platform, indicating improvement of learning and memory abilities of KO. In addition, Aβ_25–35_-injected control mice showed shorter latency to the target area where the hidden platform was placed, compared with normal mouse. On the other hand, KO-administered group showed longer latency to target area, compared with Aβ_25–35_-induced control group. In addition, the time to reach the exposed platform was not significantly different among the all groups, indicating that learning and memory abilities are not involved in swimming or visual abilities. As shown in Figure 7B, the time to reach the exposed platform was not significantly different among all groups, indicating that learning and memory abilities are not involved in swimming or visual abilities. Previous studies reported that visible platform trial did not show different escape latency between the normal group and AD mouse, indicating that the groups have similar visual capabilities among groups [34,35].

Aβ_25–35_ causes neuronal oxidative stress, which is imbalance between antioxidant and oxidant activity by over-production of ROS [36]. The ROS and RNS such as superoxide anion, hydrogen peroxide, hydroxyl radical, NO, and peroxynitrite directly or indirectly oxidized proteins, lipids, nucleic acid in the body [37]. The neuronal oxidative stress by production of ROS and RNS leads to neuroinflammation, synaptic dysfunction, neuronal apoptosis, subsequently result in memory and cognitive impairment in AD patients [38,39]. In particular, the brain tissue is specially characterized as intense mitochondrial activity, high concentration of radical-sensitive PUFAs, and accumulation of redox-active iron ions among other tissues [3,40]. Therefore, brain tissue is more vulnerable to oxidative stress, which plays a key role in the development of AD. The Aβ_25–35_ bound with active redox metal ions, which produces oxidative stress by formation of ROS, resulting in neuronal cell death [2]. In addition, the Aβ_25–35_-injected mouse increased levels of oxidative stress-related biomarkers such as ROS, MDA, and NO [9,41]. In our results, Aβ_25–35_-injected mouse significantly increased levels of ROS, MDA, and NO in the brain, compared with normal group, resulting in the neuronal oxidative stress by injection of Aβ_25–35_. However, KO and donepezil-administered mice showed significant decreases of ROS, MDA, and NO levels. In particular, the levels of ROS, MDA, and NO were decreased in the KO-administered group, similarly to the donepezil administered group. Therefore, KO and donepezil attenuated cognitive impairment by inhibition of the neuronal oxidative stress.

Apoptosis plays a critical role in the pathogenesis of various neurodegenerative diseases including AD [42]. Neuronal apoptosis is regulated by Bcl-2 family of proteins, which is the major mitochondria-mediated intrinsic apoptotic process [43]. The Bcl-2 family consists of pro-apoptotic proteins such as Bax, and anti-apoptotic proteins such as Bcl-2 [5,43]. The apoptotic stimuli such as oxidative stress and inflammation increases mitochondrial permeability and releases pro-apoptosis factors including Bax and cytochrome C, resulting in neuronal apoptosis [43,44]. Simultaneously, Bcl-2 protein inhibited an apoptotic signaling by down-regulation of Bax protein [43]. Many studies have reported that the Bcl-2 family participates in AD, especially higher ratio of Bax/Bcl-2 in the AD brain [5,45]. In addition, treatment of Aβ_25–35_ increased neuronal apoptosis by stimulation of Bax/Bcl-2 ratio in the neuronal cells and mouse brain [46,47]. In this study, Aβ_25–35_-injected control group showed significant up-regulation of pro-adpoptotic factor, Bax, and down-regulation of anti-apoptotic factor, Bcl-2, compared with normal group. However, KO-administered group decreased Bax/Bcl-2 ratio, indicating that KO attenuated neuronal apoptosis by regulation of apoptotic proteins. Therefore, KO improved cognitive function by regulation of neuronal apoptotic signaling in Aβ_25–35_-injected mouse.

## 4. Materials and Methods 

### 4.1. Materials

KO was provided by US Pharmatech Inc (Torrance, CA, USA). Aβ_25–35,_ donepezil, griess reagent, dichlorofluorescein diacetate (DCF-DA), thiobarbituric acid (TBA) and MDA were purchased from Sigma Aldrich (St. Louis, MO, USA). Sodium chloride (NaCl) was obtained from LPS Solution (Seoul, Republic of Korea). *n*-Butanol was purchased from Duksan Co. (Gyeonggi-do, Republic of Korea) and NaNO_2_ was purchased from Junsei Chimical Co. (Tokyo, Japan).

### 4.2. Animals

The ICR mice (Male, 5-weeks old, average body weight about 23–28 g) were obtained from Orient Inc. (Gyeonggi-do, Republic of Korea). The mice were maintained in a controlled humidity and temperature at 20 ± 2 °C and 50 ± 10%, respectively. The mice housed in plastic cages and allowed free access diet and water in 12 h light/dark cycle. The mice were randomly separated into six groups based on their body weights (*n* = 8 per group). Normal group was injected saline solution and orally administered water. Control group was injected Aβ_25–35_ and orally administered water. KO100, KO200 and KO500 groups were injected Aβ_25–35_ and orally administered KO at concentrations of 100, 200 and 500 mg/kg/day, respectively. Donepezil group was injected Aβ_25–35_ and orally administered donepezil at dose of 5 mg/kg/day. KO and donepezil were orally administered to mice using gastric gavage every day for 14 days. After three behavioral tests, the mice fasted for 12 h before sacrifice. The mice anesthetized with CO_2_ gas and removed the brain tissue. All experimental procedures were permitted (Approval No. PNU-2019-2465) using the guidelines established by the Pusan National University Institutional Animal Care and Use Committee (PNU-IACUC). The experimental schedule is shown in Figure 8.

### 4.3. Injection of Aβ_25–35_

For aggregation of Aβ_25–35_, Aβ_25–35_ was dissolved in saline solution and it was incubated at 37 °C for 3 days. Aggregated Aβ_25–35_ solution was intracerebroventricularly (*i.c.v.*) injected to mice in accordance with procedure established by Laursen and Belknap [48]. The mice were anesthetized with a mixture of zoletil50 and rompun (3:1 ratio) by intraperitoneal injection. After anesthetization of mouse, aggregated Aβ_25–35_ (25 nM/5 μL) was injected in the bregma of mouse into the lateral ventricle using a microinfusion pump (Baoding Longer Precision Pump Co., Ltd., Baoding, China) at the following coordinates: anterior/posterior −0.8 mm, medial/lateral −1.5 mm, and dorsal/ventral −2.2 mm [20].

### 4.4. Behavior Tests

#### 4.4.1. Novel Object Recognition Test

The novel object recognition test was carried out according to the method of Bevins and Besheer [49]. In their training day, the mice were allowed to explore two identical objects, which have the same shape and size (A, A’) in a square open-field shaped black box (40 × 30 × 20 cm) during 10 min. After 24 h, on the test day, one of the two objects was replaced by a new object (A, B) and then, the mouse was allowed to explore the two different objects, which are the familiar (A) and novel (B) object during 10 min. The number of explorations of the familiar and novel object was recorded. The object recognition ability (%) was calculated from the ratio of new object touches to the total number of familiar and novel object touches, multiplied by 100.

#### 4.4.2. Passive Avoidance Test

The passive avoidance test was performed according to the method of Newman and Kosson [32]. The passive avoidance box consisted of two chambers which have a brightly and a dark chambers with a retractable door. The dark chamber is equipped with an electric shock. In training day, mouse was placed in the light chamber of passive avoidance box. When the mouse completely entered the dark chamber, the door is closed at same time, and the mouse immediately received an electric shock (0.5 mA) for 3 s. After 24 h, in the test day, the mouse was placed into the same field and allowed to freely explore the passive avoidance box for 5 min. The time latency for the mouse to enter the dark chamber was measured in training and test days.

#### 4.4.3. Morris Water Maze Test

The Morris water maze test was conducted according to the method of Morris [50]. The Morris water maze test was carried out in a metal circular pool (80 cm diameter and 40 cm height), which is equally divided into four quadrants. Water was kept at a temperature of 22 ± 1 °C and colored with a nontoxic white poster color to hide a platform (8 cm in diameter) placed 1 cm below the surface of water in the center fixed position of one quadrant. Four visual cues were located in the walls of the apparatus for navigation. During the training periods, the hidden test consisted of three trials per day for three days. The mouse was placed into a water pool in a quadrant with a non-placed platform and allowed to find the hidden platform for 60 s. If the mouse did not find the platform within 60 s, it was gently guided to the platform for 15 s to help it remember the location of platform at end of each trial. On the test day, the mouse was allowed to find the hidden platform for 60 s as before in the primary test, and the latency time to reach the hidden platform was recorded. In a secondary test, the platform was removed and probe trials were performed. The percentage of time spent in the quadrant that previously contained the hidden platform was recorded and calculated. In a final test, the water was made transparent and the time to reach the exposed platform was measured.

### 4.5. Production of Reactive Oxygen Species (ROS)

The production of ROS in the brain was measured according to the DCF-DA assay [51]. The whole brains were homogenized with ice-cold saline solution by homogenizer (Next Advance Inc., Averill Park, NY, USA) to obtain 10% homogenate (*w*/*v*). The supernatant (10 μL) was mixed with phosphate buffered saline, and this mixture was added to 12.5 mM DCF-DA solution during 10 min. The fluorescence level of each mixture was measured at an excitation wavelength of 485 nm and an emission wavelength of 535 nm using a fluorescence spectrophotometer (FLUOstar OPTIMA, BMG Labtech, Ortenberg, Germany).

### 4.6. Lipid Peroxidation

The level of lipid peroxidation in the brain was determined by the TBARS assay [52]. A supernatant of homogenized brain tissue (200 μL) was mixed with 1% phosphoric acid and 0.67% TBA solution. This mixture was boiled for 45 min, and then cooled on ice. *n*-Butanol was added to this mixture, and it was centrifuged at 3000 rpm for 10 min. The absorbance of the supernatant was detected at 540 nm using microplate reader (Thermo Fisher Scientific, Vantaa, Finland). The level of lipid peroxidation was calculated in terms of MDA contents using the standard curve of MDA.

### 4.7. Nitric Oxide (NO) Scavenging Activity

The NO scavenging activity in the brain was determined using the method described by Schmidt et al. [53]. Brain tissue homogenate (50 μL) was mixed with distilled water. This mixture was added to the same amount of griess reagent, and then incubated at 37 °C for 30 min. The absorbance of the mixture was measured at 540 nm using microplate reader (Thermo Fisher Scientific). The concentration of NO was calculated with the standard curve of NaNO_2_ content.

### 4.8. Protein Expressions

The protein expressions in the brain were measured by the western blotting analysis. The brain tissue was lysed in radioimmunoprecipitation (RIPA) lysis buffer containing protease inhibitor cocktail at 4 °C for 30 min. The mixture was centrifuged at 12,000 rpm for 30 min, and the supernatants of mixture were slightly collected. The concentration of protein was determined Bio-Rad protein assay kit (Bio-Rad, Hercules, CA, USA) according to the Bradford method [54]. Equal amounts of proteins (15 μg) were separated on sodium dodecyl sulfate-polyacrylamide gel electrophoresis, and then transferred onto a polyvinylidene fluoride membrane. The membranes were incubated with 5% skim milk during 1 h, and were reacted with the primary antibodies such as Bax (sc-493, Santa Cruz Biotechnology Inc., Dallas, TX, USA), Bcl-2 (sc-492, Santa Cruz Biotechnology Inc.), and β-actin (#8457, Cell Signaling Technology Inc., Danvers, MA, USA) at 4 °C. After overnight, each membrane was incubated with secondary antibody at room temperature for 1 h. The protein bands were activated with enhanced chemiluminescence solution, and it was visualized with Davinch-chemi^TM^ Chemiluminescence Imaging System (CoreBio, Seoul, Republic of Korea). The density of bands was quantified with Image J software (National Institutes of Health, Bethesda, MD, USA).

### 4.9. Statistical Analysis

The results were expressed as the means ± standard deviation (SD). One-way analysis of variance (ANOVA) followed by Duncan’s multiple test were performed using SPSS software (SPSS Inc, Chicago, IL, USA). In the novel object recognition test, to evaluate comparison of familiar and novel route, Student’s *t*-test was carried out. Statistical significance was considered as *p* < 0.05.

## 5. Conclusions

In this study, we demonstrated that KO attenuated the cognitive impairment induced by Aβ_25–35_ in mice. In addition, KO inhibited oxidative stress by reduction of ROS, MDA, and NO, and apoptosis via regulation of Bax/Bcl-2 ratio in the Aβ_25–35_-induced brain tissue. Therefore, we suggest the potential of KO for the treatment and prevention of AD.

## Figures and Tables

**Figure 1 molecules-25-03942-f001:**
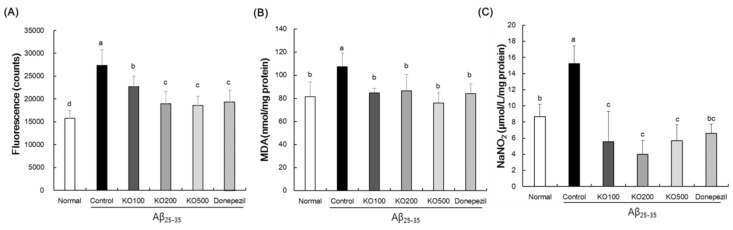
Effects of krill oil (KO) on oxidative stress-related factors such as reactive oxygen species (**A**), lipid peroxidation (**B**), and nitric oxide (**C**) in the brain. Normal, injection of saline solution + oral administration of water; Control, injection of Aβ_25–35_ + oral administration of water; KO100, injection of Aβ_25–35_ + oral administration of KO at 100 mg/kg/day; KO200, injection of Aβ_25–35_ + oral administration of KO at 200 mg/kg/day; KO500, injection of Aβ_25–35_ + oral administration of KO at 500 mg/kg/day; Donepezil, injection of Aβ_25–35_ + oral administration of donepezil at 5 mg/kg/day. Values are represented as means ± SD (*n* = 8). ^a–d^ Means with the different letters are significantly different (*p* < 0.05) by Duncan’s multiple range test. ^b,c^ Means with the same letter such as b and c are not significantly different (*p* < 0.05) by Duncan’s multiple range test.

**Figure 2 molecules-25-03942-f002:**
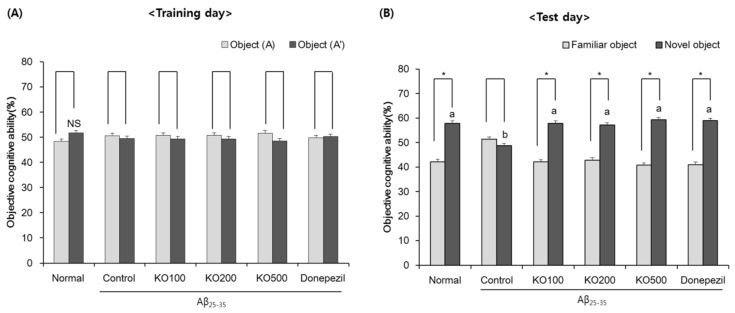
Novel object recognition test. (**A**) The percentage of time exploring of the identical two objects in the training day. (**B**) The percentage of time exploring of the familiar and novel object in the test day. Normal, injection of saline solution + oral administration of water; Control, injection of Aβ_25–35_ + oral administration of water; KO100, injection of Aβ_25–35_ + oral administration of KO at 100 mg/kg/day; KO200, injection of Aβ_25–35_ + oral administration of KO at 200 mg/kg/day; KO500, injection of Aβ_25–35_ + oral administration of KO at 500 mg/kg/day; Donepezil, injection of Aβ_25–35_ + oral administration of donepezil at 5 mg/kg/day. Values are represented as means ± SD (*n* = 8). * The object cognitive abilities for familiar and novel objects are significantly different as determined by Student’s *t*-test (*p* < 0.05). ^a,b^ Means with the different letters are significantly different (*p* < 0.05) by Duncan’s multiple range test. NS, non-significance.

**Figure 3 molecules-25-03942-f003:**
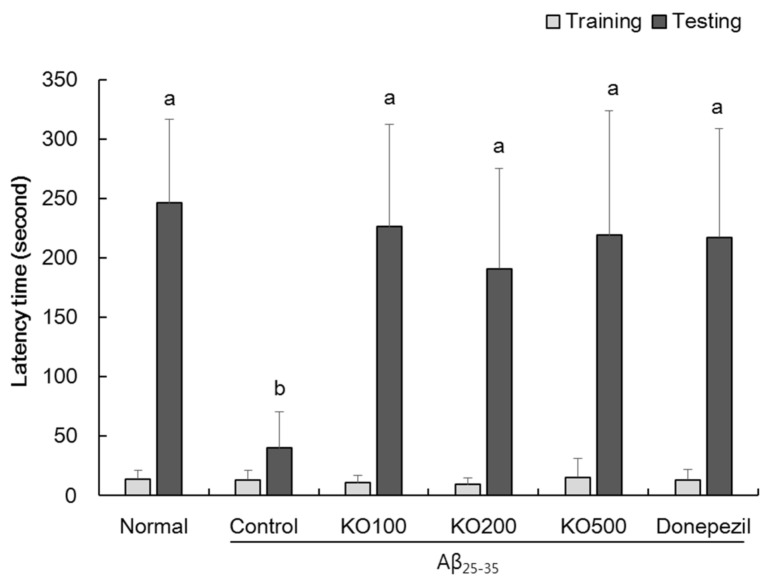
Effects of krill oil (KO) on passive avoidance ability in the passive avoidance test. Normal, injection of saline solution + oral administration of water; Control, injection of Aβ_25–35_ + oral administration of water; KO100, injection of Aβ_25–35_ + oral administration of KO at 100 mg/kg/day; KO200, injection of Aβ_25–35_ + oral administration of KO at 200 mg/kg/day; KO500, injection of Aβ_25–35_ + oral administration of KO at 500 mg/kg/day; Donepezil, injection of Aβ_25–35_ + oral administration of donepezil at 5 mg/kg/day. Values are represented as means ± SD (*n* = 8). ^a,b^ Means with the different letters are significantly different (*p* < 0.05) by Duncan’s multiple range test.

**Figure 4 molecules-25-03942-f004:**
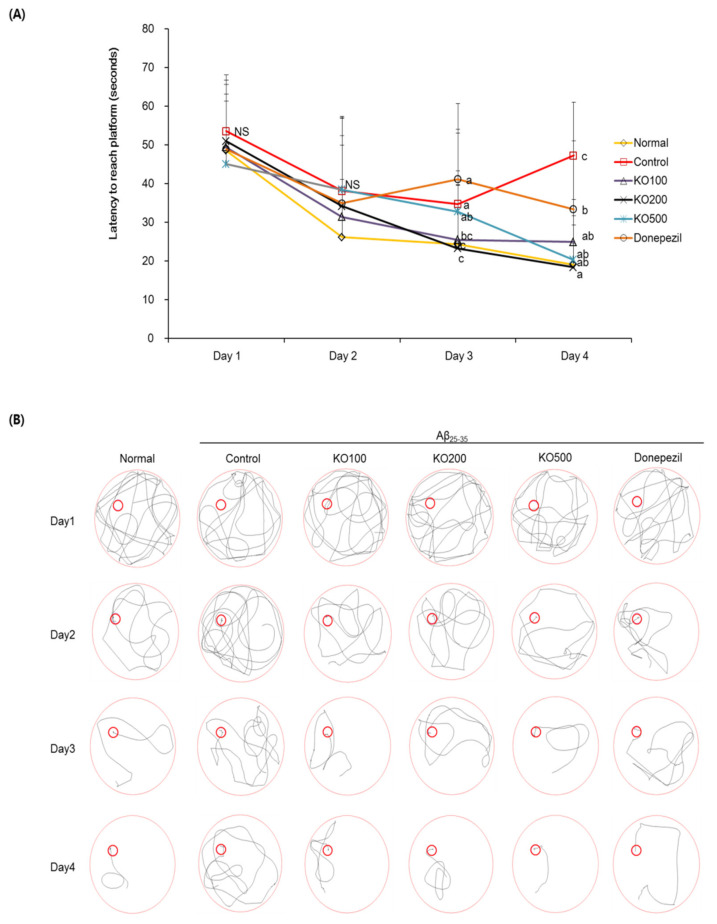
Effects of krill oil (KO) on escape latency to the hidden platform in the Morris water maze test (**A**). Path tracing of each groups in the Morris water maze test (**B**). Normal, injection of saline solution + oral administration of water; Control, injection of Aβ_25–35_ + oral administration of water; KO100, injection of Aβ_25–35_ + oral administration of KO at 100 mg/kg/day; KO200, injection of Aβ_25–35_ + oral administration of KO at 200 mg/kg/day; KO500, injection of Aβ_25–35_ + oral administration of KO at 500 mg/kg/day; Donepezil, injection of Aβ_25–35_ + oral administration of donepezil at 5 mg/kg/day. Values are represented as means ± SD (*n* = 8). ^a–c^ Means with the different letters are significantly different (*p* < 0.05) by Duncan’s multiple range test. NS, non-significance. ^a,b^ Means with the same letters such as a and b are not significantly different (*p* < 0.05) by Duncan’s multiple range test.

**Figure 5 molecules-25-03942-f005:**
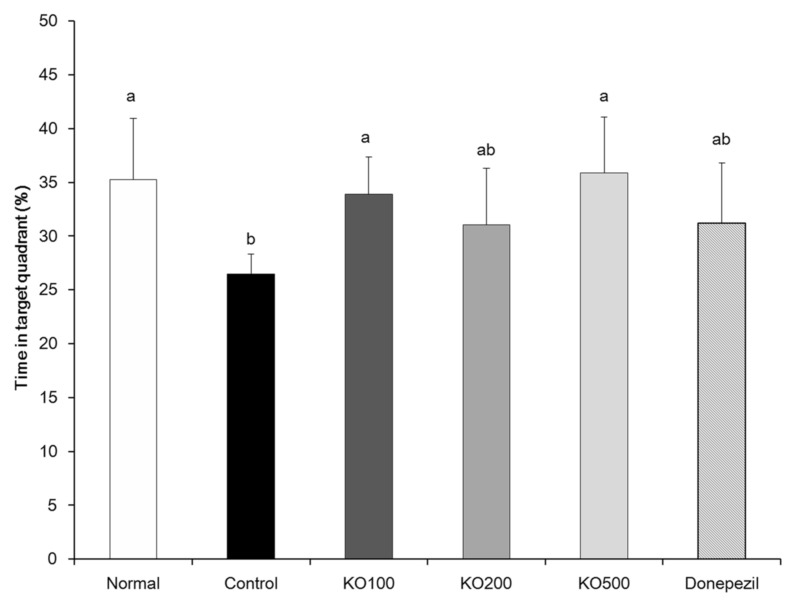
Effects of krill oil (KO) on learning and memory impairments in the Morris water maze test. Normal, injection of saline solution + oral administration of water; Control, injection of Aβ_25–35_ + oral administration of water; KO100, injection of Aβ_25–35_ + oral administration of KO at 100 mg/kg/day; KO200, injection of Aβ_25–35_ + oral administration of KO at 200 mg/kg/day; KO500, injection of Aβ_25–35_ + oral administration of KO at 500 mg/kg/day; Donepezil, injection of Aβ_25–35_ + oral administration of donepezil at 5 mg/kg/day. Values are represented as means ± SD (*n* = 8). ^a,b^ Means with the different letters are significantly different (*p* < 0.05) by Duncan’s multiple range test. ^a,b^ Means with the same letters such as a and b are not significantly different (*p* < 0.05) by Duncan’s multiple range test.

**Figure 6 molecules-25-03942-f006:**
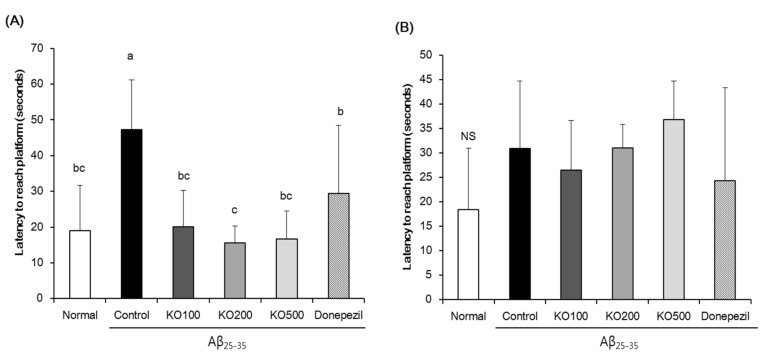
Effect of krill oil (KO) on on escape latency to the hidden platform (**A**) and exposed platfrom (**B**) in Morris water maze test. Normal, injection of saline solution + oral administration of water; Control, injection of Aβ_25–35_ + oral administration of water; KO100, injection of Aβ_25–35_ + oral administration of KO at 100 mg/kg/day; KO200, injection of Aβ_25–35_ + oral administration of KO at 200 mg/kg/day; KO500, injection of Aβ_25–35_ + oral administration of KO at 500 mg/kg/day; Donepezil, injection of Aβ_25–35_ + oral administration of donepezil at 5 mg/kg/day. Values are represented as means ± SD. ^a–c^ Means with the different letters are significantly different (*p* < 0.05) by Duncan’s multiple range test (*n* = 8). ^b,c^ Means with the same letters such as b and c are not significantly different (*p* < 0.05) by Duncan’s multiple range test. NS, non-significance.

**Figure 7 molecules-25-03942-f007:**
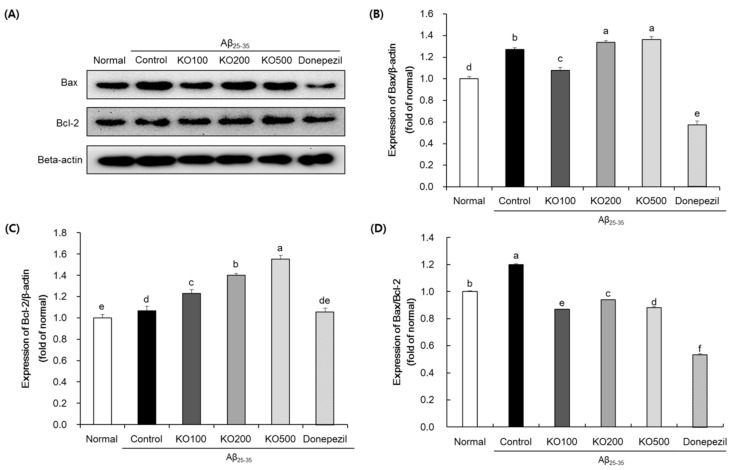
Effects of krill oil (KO) on Bax and Bcl-2 protein expressions in the brain. Western blot band (**A**), Bax (**B**), Bcl-2 (**C**), and Bax/Bcl-2 ratio (**D**). Normal, injection of saline solution + oral administration of water; Control, injection of Aβ_25–35_ + oral administration of water; KO100, injection of Aβ_25–35_ + oral administration of KO at 100 mg/kg/day; KO200, injection of Aβ_25–35_ + oral administration of KO at 200 mg/kg/day; KO500, injection of Aβ_25–35_ + oral administration of KO at 500 mg/kg/day; Donepezil, injection of Aβ_25–35_ + oral administration of donepezil at 5 mg/kg/day. Values are represented as means ± SD (*n* = 8). ^a–f^ Means with the different letters are significantly different (*p* < 0.05) by Duncan’s multiple range test. ^d,e^ Means with the same letters such as d and e are not significantly different (*p* < 0.05) by Duncan’s multiple range test. β-actin was used as a loading control.

**Figure 8 molecules-25-03942-f008:**
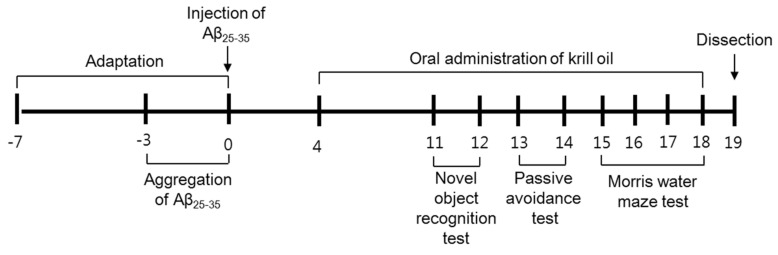
Experimental schedule.

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
