# Peer review of "Krill Oil Attenuates Cognitive Impairment by the Regulation of Oxidative Stress and Neuronal Apoptosis in an Amyloid β-Induced Alzheimer’s Disease Mouse Model"

_molecules, 2020, doi:10.3390/molecules25173942_

Round 1

Reviewer 1 Report

Kim et al. investigate the effect of krill oil on cognitive improvements in an Ab(25-35) induced AD mouse model. The manuscript is technically sound and provides insight into possible therapeutic value of naturally derived products against AD. Few minor issues need to be addressed:

L 347-350: This conclusion is not supported by literature. In fact Morris showed that spatial memory is very important during the search for the platform (doi: 10.1038/13202). Authors should discuss other possible theories as to why the treatment showed no significant results. Fig 7 results can be used as a starting point.

Show representative swim paths/trajectories for the water maze for each of the groups.

Fig 2 legend is incorrect: spatial perception is not tested, thus there are no routes but objects.

Author Response

Thank you for the valuable comments on this paper. We considered the comments carefully and the manuscript has been revised according to the comments.

Reviewer #1: Kim et al. investigate the effect of krill oil on cognitive improvements in an Ab(25-35) induced AD mouse model. The manuscript is technically sound and provides insight into possible therapeutic value of naturally derived products against AD. Few minor issues need to be addressed:

L 347-350: This conclusion is not supported by literature. In fact Morris showed that spatial memory is very important during the search for the platform (doi: 10.1038/13202). Authors should discuss other possible theories as to why the treatment showed no significant results. Fig 7 results can be used as a starting point.

; According to the comment, it was discussed with other related studies. As shown in Fig 6B, the time to reach the exposed platform was not significantly different among all groups, indicating that learning and memory abilities are not involved in swimming or visual abilities. Previous studies reported that visible platform trial did not show different escape latency between the normal group and Alzheimer’s disease mouse, indicating that the groups have similar visual capabilities among groups (Bromley-Brits et al., 2011; Choi et al., 2013). We added it in Discussion section (Line 375-380, Page 11). In addition, Figure 7 revised to Figure 2 for use as a starting point in revised manuscript.

[References]

Bromley-Brits, K.; Deng, Y.; Song, W. Morris water maze test for learning and memory deficits in Alzheimer's disease model mice. J Vis Exp 2011, 53, 2920.

Choi, J.Y.; Cho, E.J.; Lee, H.S.; Lee, J.M.; Yoon, Y.H.; Lee, S. Tartary buckwheat improves cognition and memory function in an in vivo amyloid-β-induced Alzheimer model. Food Chem Toxicol 2013, 53, 105-111.

Show representative swim paths/trajectories for the water maze for each of the groups.

; We added the swim paths for the water maze for each of the groups in Figure 5B.

Figure 5. Effects of krill oil (KO) on escape latency to the hidden platform in the Morris water maze test (A). Path tracing of each groups in the Morris water maze test (B). Normal, injection of saline solution + oral administration of water; Control, injection of Aβ25-35 + oral administration of water; KO100, injection of Aβ25-35 + oral administration of KO at 100 mg/kg/day; KO200, injection of Aβ25-35 + oral administration of KO at 200 mg/kg/day; KO500, injection of Aβ25-35 + oral administration of KO at 500 mg/kg/day; Donepezil, injection of Aβ25-35 + oral administration of donepezil at 5 mg/kg/day. Values are represented as means ± SD (n = 8). a-cMeans with the different letters are significantly different (P < 0.05) by Duncan’s multiple range test. NS, non-significance.

Fig 2 legend is incorrect: spatial perception is not tested, thus there are no routes but objects.

; We revised Fig 2 legend (Fig 3 in revised manuscript).

[Fig 3 legend]

*The object cognitive abilities for familiar and novel objects are significantly different as determined by Student’s t-test (P < 0.05).

Reviewer 2 Report

  1. this is a very interesting study since in parallel they tested a drug already being in the therapeutic armamentarium for Alz disease.
  2. the study is very well designed and
  3. 3. the results are amazing. In addition, it is well written.

Author Response

Thank you for your attention.

Reviewer 3 Report

The manuscript submitted to Molecules by Kim-JH et al. evaluated the efficacy of krill oil (daily gavage for 14 days) to attenuate cognitive deficits induced by i.c.v. injection of Aβ(25-35) peptides.  The authors report that animals i.c.v. injected with aggregated Aβ(25-35) peptides had statistically significant performance deficits in all three of their tested behavioral tasks: novel object recognition, passive avoidance, and Morris water maze (spaced training).  Aβ(25-35) peptide-injected animals that also received treatment with krill oil were found to have significantly improved behavioral performances in all tasks, comparable to the positive control treatment (Donepezil, an FDA-approved treatment for Alzheimer’s Disease). Further, after behavioral testing, brain tissues were harvested and examined for evidence of oxidative damage and potential apoptotic response.  Similar to the behavioral test results, the authors report evidence of increased oxidative stress (lipid peroxidation, increased ROS and NO production) and apoptosis (Bax/Bcl-2 ratio) after Aβ(25-35) peptide injection that was attenuated by treatment with either krill oil or Donepezil.  Altogether, the manuscript presents experimental evidence that krill oil, a complex mixture of compounds including multiple polyunsaturated fats and natural antioxidants, may have beneficial effects at limiting ROS-mediated damage and cognitive deficits in a non-genetic model of experimental AD.  Overall results are potentially interesting and the manuscript is well written (though some passages that could be made more clear, e.g. lines 78-79 could be interpreted to mean than animals were segregated by weight-smaller with smaller, larger with larger).  However, in my estimation, some aspects of the data are not particularly compelling and need to be reexamined.

  • For NOR (figure 2), were the animals familiarized to the arena first? What was their performance on the objects during the training phase?  50-50?  Did the groups all explore/interact equivalently?
  • The Morris water maze quadrant preference data (figure 5) shows between approximately 18%- 25% occupancy in the target (platform) quadrant. That is basically at or below chance level.  Quadrant preference should be analyzed within a group to first show that the animals did indeed have a target quadrant preference.  Post hoc you can compare between groups.  Compared to the probe times, the visible platform test produced what I would characterize as relatively long latency times.  Can this be explained?
  • If tracking software was employed, the MWM task provides many additional metrics that could be examined besides just latency to platform and quadrant preference. Swim path, speed, distance traveled, and platform crossings are a few commonly reported metrics. 
  • Were the MWM training curves statistically different between groups? This is the only learning metric you presented, the rest are memory related.
  • The representative western blot images presented in the top panel of figure 8 do not match the group data presented in the bottom panel. The control lane appears to be well above a 2-fold change, and to me the KO200 and KO500 lanes also appear to be >1. Were data normalized to beta-actin, or is that just shown to verify loading?   What was the N examined?  Are the standard deviations really so small as to be not visible?
  • The Methods section is unclear and/or missing critical details in several sections. Line 96: What precisely does 25 nM/ 5 µL mean?  I assume it means you injected 5 µL of a 25 nM solution?  Over how long?  For a mouse brain, 5 µL is a large volume.  Was the 25 nM concentration based on the calculated initial monomeric peptide concentration, or was the concentration determined after aggregation (dynamic light scattering or related method)?  Line 137: How were the brains homogenized?  What region(s)?  What volumes and concentrations of brain homogenate were used in the ROS/LP/NO assays.  Line 162: equal amounts is how much protein?  Line 164: Primary and secondary antibody information (catalog number, concentration used).  Check your descriptions of the passive avoidance task in both the methods and results sections.  Sometimes it is presented as the shock occurring in the lighted area rather than the darkened area.
  • I do not understand the use of the letters a, b, c in the figures to denote statistical significance. Only one P-level is given in the legend.  What does a, b and c indicate beyond meeting a P<0.05 threshold?  Including group sizes would be helpful.

Author Response

Thank you for the valuable comments on this paper. We considered the comments carefully and the manuscript has been revised according to the comments.

Reviewer #2: The manuscript submitted to Molecules by Kim-JH et al. evaluated the efficacy of krill oil (daily gavage for 14 days) to attenuate cognitive deficits induced by i.c.v. injection of Aβ(25-35) peptides. The authors report that animals i.c.v. injected with aggregated Aβ(25-35) peptides had statistically significant performance deficits in all three of their tested behavioral tasks: novel object recognition, passive avoidance, and Morris water maze (spaced training). Aβ(25-35) peptide-injected animals that also received treatment with krill oil were found to have significantly improved behavioral performances in all tasks, comparable to the positive control treatment (Donepezil, an FDA-approved treatment for Alzheimer’s Disease). Further, after behavioral testing, brain tissues were harvested and examined for evidence of oxidative damage and potential apoptotic response. Similar to the behavioral test results, the authors report evidence of increased oxidative stress (lipid peroxidation, increased ROS and NO production) and apoptosis (Bax/Bcl-2 ratio) after Aβ(25-35) peptide injection that was attenuated by treatment with either krill oil or Donepezil. Altogether, the manuscript presents experimental evidence that krill oil, a complex mixture of compounds including multiple polyunsaturated fats and natural antioxidants, may have beneficial effects at limiting ROS-mediated damage and cognitive deficits in a non-genetic model of experimental AD. Overall results are potentially interesting and the manuscript is well written (though some passages that could be made more clear, e.g. lines 78-79 could be interpreted to mean than animals were segregated by weight-smaller with smaller, larger with larger). However, in my estimation, some aspects of the data are not particularly compelling and need to be reexamined.

For NOR (figure 2), were the animals familiarized to the arena first? What was their performance on the objects during the training phase? 50-50? Did the groups all explore/interact equivalently?

; On the first day (training day), all experimental animals freely explored two identical objects (A, A’). All groups did not show significant difference of the percentage of time exploring of between two identical objects (about 50-50), and equivalently explored objects (Supplemental Figure 1A). In the test day, one of the two objects was replaced with a novel object (A, B). The Aβ25-35-non-injected normal group showed higher exploration of novel object than that of the familiar object, while Aβ25-35-injected control group showed non-significance between familiar and novel objects (Figure 2). We added it in results section (Line 212-219, Page 6).

Figure 2. Novel object recognition test. (A) The percentage of time exploring of the identical two objects in the training day. (B) The percentage of time exploring of the familiar and novel object in the test day. Normal, injection of saline solution + oral administration of water; Control, injection of Aβ25-35 + oral administration of water; KO100, injection of Aβ25-35 + oral administration of KO at 100 mg/kg/day; KO200, injection of Aβ25-35 + oral administration of KO at 200 mg/kg/day; KO500, injection of Aβ25-35 + oral administration of KO at 500 mg/kg/day; Donepezil, injection of Aβ25-35 + oral administration of donepezil at 5 mg/kg/day. Values are represented as means ± SD. *The object cognitive abilities for familiar and novel objects are significantly different as determined by Student’s t-test (P < 0.05). a-bMeans with the different letters are significantly different (P < 0.05) by Duncan’s multiple range test. NS, non-significance.

The Morris water maze quadrant preference data (figure 5) shows between approximately 18%- 25% occupancy in the target (platform) quadrant. That is basically at or below chance level.

; In original manuscript (Figure 5), occupancy in target quadrant (%) of the experimental groups in figures was calculated relative to the normal group 25.00% in the Morris water maze test. However, according to the comment, the result (Fig. 6 in revised manuscript) was re-analyzed and expressed the % time spent in the platform-located target quadrant according to other previous studies. The normal group stayed target quadrant during 35.26 ± 5.70% (21.16 s of 60 s), while control group stayed target quadrant during 26.47 ± 1.84% (15.88 s of 60 s). In addition, KO100, KO200, KO500, and donepezil groups showed the percentage of target quadrant without non-calculated relative normal group, 33.87 ± 3.46%, 31.05 ± 5.26%, 35.90 ± 5.18%, and 31.21 ± 5.61%, respectively. Therefore, the experimental groups performed above chance level. We added it in results section (Line 263-268, Page 7)

Figure 6. Effects of krill oil (KO) on learning and memory impairments in the Morris water maze test. Normal, injection of saline solution + oral administration of water; Control, injection of Aβ25-35 + oral administration of water; KO100, injection of Aβ25-35 + oral administration of KO at 100 mg/kg/day; KO200, injection of Aβ25-35 + oral administration of KO at 200 mg/kg/day; KO500, injection of Aβ25-35 + oral administration of KO at 500 mg/kg/day; Donepezil, injection of Aβ25-35 + oral administration of donepezil at 5 mg/kg/day. Values are represented as means ± SD (n = 8). a-bMeans with the different letters are significantly different (P < 0.05) by Duncan’s multiple range test.

[References]

Riedel G, Micheau J, Lam AG, et al. Reversible neural inactivation reveals hippocampal participation in several memory processes. Nat Neurosci. 1999;2(10):898-905. doi:10.1038/13202

Yan XS, Yang ZJ, Jia JX, et al. Protective mechanism of testosterone on cognitive impairment in a rat model of Alzheimer's disease. Neural Regen Res. 2019;14(4):649-657. doi:10.4103/1673-5374.245477

Quadrant preference should be analyzed within a group to first show that the animals did indeed have a target quadrant preference. Post hoc you can compare between groups.

; Quadrant preference (Fig. 5 in original manuscript = Fig. 6 in revised manuscript) was subjected to one-way analysis of variance (ANOVA) and Duncan’s post hoc test (P < 0.05). The different letters mean significant difference, while same letters mean non-significant difference among groups. As shown in Fig. 6, normal group (a) and control group (b) showed significant difference (P < 0.05) by different letters among groups in Duncan’s post hoc test. However, normal group (a) and KO100 group (a) showed non-significance (P < 0.05) by same letters among groups. In addition, similarly to our study, many previous studies have used Duncan’s post hoc test to analyze results (Liu et al., 2012; Xia et al., 2015).

[References]

Liu, S.H.; Chang, C.D.; Chen, P.H.; Su, J.R.; Chen, C.C.; Chaung, H.C. Docosahexaenoic acid and phosphatidylserine supplementations improve antioxidant activities and cognitive functions of the developing brain on pentylenetetrazol-induced seizure model. Brain Research, 2012, 1451, 19-26.

Xia, S.F.; Xie, Z.X.; Qiao, Y, et al. Differential effects of quercetin on hippocampus-dependent learning and memory in mice fed with different diets related with oxidative stress. Physiol Behav 2015, 138, 325-331.

Compared to the probe times, the visible platform test produced what I would characterize as relatively long latency times. Can this be explained?

; In the visible platform test, the mouse found the visible platform without visible cues. However, in the probe times, mouse found the hidden platform with visible cues by learning and memory.

If tracking software was employed, the MWM task provides many additional metrics that could be examined besides just latency to platform and quadrant preference. Swim path, speed, distance traveled, and platform crossings are a few commonly reported metrics.

; We performed the Morris water maze test using tracking software and explained the latency to platform and quadrant preference for examine the learning and memory abilities. In addition, we added the effects of swim path in Figure 5B. Further study will need to the effects of swim speed, distance traveled, and platform crossings in Morris water maze test.

Figure 5. Effects of krill oil (KO) on escape latency to the hidden platform in the Morris water maze test (A). Path tracing of each groups in the Morris water maze test (B). Normal, injection of saline solution + oral administration of water; Control, injection of Aβ25-35 + oral administration of water; KO100, injection of Aβ25-35 + oral administration of KO at 100 mg/kg/day; KO200, injection of Aβ25-35 + oral administration of KO at 200 mg/kg/day; KO500, injection of Aβ25-35 + oral administration of KO at 500 mg/kg/day; Donepezil, injection of Aβ25-35 + oral administration of donepezil at 5 mg/kg/day. Values are represented as means ± SD (n = 8). a-cMeans with the different letters are significantly different (P < 0.05) by Duncan’s multiple range test. NS, non-significance.

Were the MWM training curves statistically different between groups? This is the only learning metric you presented, the rest are memory related.

; We additionally statistical analysis in MWM training curves in Figure 5A.

Figure 5. Effects of krill oil (KO) on escape latency to the hidden platform in the Morris water maze test (A). Path tracing of each groups in the Morris water maze test (B). Normal, injection of saline solution + oral administration of water; Control, injection of Aβ25-35 + oral administration of water; KO100, injection of Aβ25-35 + oral administration of KO at 100 mg/kg/day; KO200, injection of Aβ25-35 + oral administration of KO at 200 mg/kg/day; KO500, injection of Aβ25-35 + oral administration of KO at 500 mg/kg/day; Donepezil, injection of Aβ25-35 + oral administration of donepezil at 5 mg/kg/day. Values are represented as means ± SD (n = 8). a-cMeans with the different letters are significantly different (P < 0.05) by Duncan’s multiple range test. NS, non-significance.

The representative western blot images presented in the top panel of figure 8 do not match the group data presented in the bottom panel. The control lane appears to be well above a 2-fold change, and to me the KO200 and KO500 lanes also appear to be >1. Were data normalized to beta-actin, or is that just shown to verify loading?   What was the N examined? Are the standard deviations really so small as to be not visible?

; The protein expressions of Bax/Bcl-2 ratio normalized to beta-actin in Figure 8. We additionally expressed Bax (Fig. 8B) and Bcl-2 (Fig. 8C) by normalization of beta-actin, respectively. In addition, we added group sizes (n = 8) in Figure legends. The standard deviations are rechecked.

Figure 8. Effects of krill oil (KO) on Bax and Bcl-2 protein expressions in the brain. Western blot band (A), Bax (B), Bcl-2 (C), and Bax/Bcl-2 ratio (D). Normal, injection of saline solution + oral administration of water; Control, injection of Aβ25-35 + oral administration of water; KO100, injection of Aβ25-35 + oral administration of KO at 100 mg/kg/day; KO200, injection of Aβ25-35 + oral administration of KO at 200 mg/kg/day; KO500, injection of Aβ25-35 + oral administration of KO at 500 mg/kg/day; Donepezil, injection of Aβ25-35 + oral administration of donepezil at 5 mg/kg/day. Values are represented as means ± SD. a-fMeans with the different letters are significantly different (P < 0.05) by Duncan’s multiple range test. Beta-actin was used as a loading control.

The Methods section is unclear and/or missing critical details in several sections.

Line 96: What precisely does 25 nM/ 5 µL mean? I assume it means you injected 5 µL of a 25 nM solution? Over how long? For a mouse brain, 5 µL is a large volume. Was the 25 nM concentration based on the calculated initial monomeric peptide concentration, or was the concentration determined after aggregation (dynamic light scattering or related method)?

; According to the procedure outlined by previous studies (Choi et al., 2014; Maurice et al., 1996; Haley and Mccormick, 1957), to induce Alzheimer’s disease mouse model, 25 nM/ 5 µL of monomeric amyloid beta peptide solution was injected mouse (5 μL /per mouse) at a rate of 1 μL/min. The 25 nM concentration was calculated on the basis of the concentration of initial monomeric peptide.

[References]

Choi, Y.Y.; Maeda, T.; Fujii, H, et al. Oligonol improves memory and cognition under an amyloid β(25-35)-induced Alzheimer's mouse model. Nutr Res 2014, 34, 595-603.

Maurice, T.; Lockhart, B.P.; Privat, A. Amnesia induced in mice by centrally administered beta-amyloid peptides involves cholinergic dysfunction. Brain Res 1996, 706, 181-193.

Haley, T.J.; Mccormick, W.G. Pharmacological effects produced by intracerebral injection of drugs in the conscious mouse. Br J Pharmacol Chemother 1957, 12, 12-15.

Line 137: How were the brains homogenized? What region(s)? What volumes and concentrations of brain homogenate were used in the ROS/LP/NO assays.

; The whole brains were homogenized with ice-cold saline solution by homogenizer (Next Advance Inc., Averill Park, NY, USA) to obtain 10% homogenate (w/v) (Driver et al., 2000). The volumes of brain homogenate used 10, 200, 50 μL in the ROS, MDA, and NO assays, respectively. We added it in Materials and Methods section (Line 136-152, Page 4)

[References]

Driver, A.S.; Kodavanti, P.R.; Mundy, W.R. Age-related changes in reactive oxygen species production in rat brain homogenates. Neurotoxicol Teratol, 2000, 22, 175-181.

Line 162: equal amounts is how much protein?

; We revised it in Materials and Methods (Line 163, Page 4).

[Materials and Methods]

Equal amounts of proteins (15 ug) were separated were separated on sodium dodecyl sulfate-polyacrylamide gel electrophoresis, and then transferred onto a polyvinylidene fluoride membrane.

Line 164: Primary and secondary antibody information (catalog number, concentration used).

; We added it in Materials and Methods (Line 166-168, Page 4).

[Materials and Methods]

The primary antibodies such as Bax (sc-493, Santa Cruz Biotechnology Inc., Dallas, Texas, USA), Bcl-2 (sc-492, Santa Cruz Biotechnology Inc., Dallas, Texas, USA), and β-actin (#8457, Cell Signaling Technology Inc., Danvers, MA, USA).

Check your descriptions of the passive avoidance task in both the methods and results sections. Sometimes it is presented as the shock occurring in the lighted area rather than the darkened area.

; We revised descriptions of passive avoidance task (Line 118, Page 5).

[Materials and Methods]

The time latency for the mouse to enter the dark chamber was measured in training and test days.

I do not understand the use of the letters a, b, c in the figures to denote statistical significance. Only one P-level is given in the legend. What does a, b and c indicate beyond meeting a P<0.05 threshold? Including group sizes would be helpful.

; The all results subjected to one-way analysis of variance (ANOVA) and Duncan’s post hoc test (P < 0.05). The different letters means significant difference, while same letters means non-significant difference among groups. As shown in Fig. 5, normal group (a) and control group (b) showed significant difference (P < 0.05) by different letters among groups in Duncan’s post hoc test. However, normal group (a) and KO100 group (a) showed non-significance (P < 0.05) by same letters among groups. In addition, similarly to our study, many previous studies have used Duncan’s post hoc test to analyze results (Liu et al., 2012; Xia et al., 2015). Furthermore, we added group sizes (n = 8) in all Figure legends.

[References]

Liu, S.H.; Chang, C.D.; Chen, P.H.; Su, J.R.; Chen, C.C.; Chaung, H.C. Docosahexaenoic acid and phosphatidylserine supplementations improve antioxidant activities and cognitive functions of the developing brain on pentylenetetrazol-induced seizure model. Brain Research, 2012, 1451, 19-26.

Xia, S.F.; Xie, Z.X.; Qiao, Y, et al. Differential effects of quercetin on hippocampus-dependent learning and memory in mice fed with different diets related with oxidative stress. Physiol Behav 2015, 138, 325-331.

Round 2

Reviewer 3 Report

The authors have addressed my previous concerns